# Evaluation of Two Tests for the Rapid Detection of CTX-M Producers Directly in Urine Samples

**DOI:** 10.3390/antibiotics12111585

**Published:** 2023-11-02

**Authors:** Forrest Tang, Chung-Ho Lee, Xin Li, Shuo Jiang, Kin-Hung Chow, Cindy Wing-Sze Tse, Pak-Leung Ho

**Affiliations:** 1Department of Clinical Pathology, Kwong Wah Hospital, Hospital Authority, Hong Kong, China; forresttang97@gmail.com (F.T.);; 2Department of Microbiology, University of Hong Kong, Hong Kong, China; 3Carol Yu Centre for Infection, University of Hong Kong, Hong Kong, China; 4Department of Microbiology, Queen Mary Hospital, Hospital Authority, Hong Kong, China

**Keywords:** beta-lactamases, CTX-M, extended-spectrum beta-lactamases, immunochromatographic assay, colorimetric assay, *Enterobacteriaceae*, urinary tract infections

## Abstract

Infections caused by extended-spectrum β-lactamase-producing *Enterobacterales* have increased rapidly and are mainly attributed to the production of CTX-M enzymes. This study evaluated the NG-Test^®^ CTX-M MULTI lateral flow assay (CTX-M LFA) and the Rapid ESBL NP^®^ test (ESBL NP test) for rapid detection of CTX-M-producing Enterobacterales directly in midstream urine (MSU) samples. Testing was performed on 277 clinical MSU samples in a hospital microbiology laboratory from November 2022 to January 2023; 60 of these samples (30 positive for ESBL producers and 30 positive for non-ESBL producers) were tested retrospectively after the identification and susceptibility results were obtained, and 217 samples were tested prospectively immediately after a Gram stain showing the presence of Gram-negative bacilli. The results were compared against phenotypic detection of ESBL and molecular testing as the reference methods. Overall, 67 of the 277 samples were culture-positive for ESBL-producing Enterobacterales. PCR for the *bla*_CTX-M_ gene was positive for all ESBL-producing Enterobacterales isolates. All CTX-M LFA results were interpretable, while three of the ESBL NP test results were noninterpretable. The sensitivity of the CTX-M LFA (100%, 95% CI 94.6–100%) was higher than that of the ESBL NP test (86.6%, 95% CI 76.0–93.7%). Both tests had high specificities (CTX-M LFA, 99.1%, 95% CI 96.6–99.9% and ESBL NP test, 100%, 95% CI 98.2–100%). In conclusion, both the CTX-M LFA and the ESBL NP test can deliver rapid results that could improve antimicrobial stewardship for urinary tract infections.

## 1. Introduction

Antimicrobial resistance due to extended-spectrum β-lactamase (ESBL)-producing Enterobacterales poses a significant threat to the global public health system [1]. Clinically, infections due to ESBL-producing Enterobacterales may lead to increased morbidity and mortality due to delays in the use of effective antibiotics, as these organisms are resistant to penicillins and broad-spectrum cephalosporins [2,3]. In many parts of the world, CTX-M-type β-lactamases account for the majority of ESBLs found clinically [4]. There are five major groups of CTX-M enzymes, namely, groups 1, 2, 8, 9 and 25 [5]. The CTX-M-15 β-lactamase, which belongs to CTX-M group 1 enzymes, is dominant in most regions globally except for Asia, where CTX-M group 9 enzymes are more prevalent [1,5]. In Hong Kong, CTX-M-producing Enterobacterales account for more than 90% of ESBL-producing Enterobacterales encountered clinically, with the majority being producers of CTX-M group 9 enzymes followed by group 1 enzymes [6,7,8].

Urinary tract infections (UTIs), both community-acquired or nosocomial, are commonly caused by members of Enterobacterales [9]. The increase in the overall prevalence of ESBL-producing Enterobacterales has caused an increase in the prevalence of UTIs due to these resistant organisms. Delay in effective antimicrobial therapy for treatment may lead to severe consequences such as sepsis, renal scarring and prolonged hospital stay [10]. Therefore, there is a need for rapid detection of ESBLs in urine samples to guide timely escalation of antimicrobial therapy.

Conventional detection of ESBLs in Enterobacterales using the combined disc test and the double disc synergy test requires isolated bacterial colonies for testing and an incubation time of 16–18 h before the results can be read. The long turnaround time may cause a delay in effective antimicrobial therapy, which may lead to adverse patient outcomes. This possibility has incentivized the development of rapid tests for the detection of ESBL production. The Rapid ESBL NP^®^ test (Liofilchem, Roseto degli Abbruzzi, Italy) (ESBL NP test) is a colorimetric test that uses the hydrolysis of cefotaxime and inhibition of the hydrolysis by tazobactam as a surrogate marker for ESBL production [11]. The NG-Test^®^ CTX-M MULTI test (NG-Biotech Laboratories, Guipry, France) is a lateral flow assay (CTX-M LFA) that uses a cocktail of monoclonal antibodies to detect all five groups of CTX-M enzymes [12]. Apart from testing isolated bacterial colonies, both tests have been evaluated for direct detection of ESBLs from positive blood culture broths with good performance [13,14]. However, data on their performance in direct detection of ESBLs in urine specimens is lacking. This study aimed to evaluate the diagnostic performance of the CTX-M LFA and the ESBL NP test for direct detection of ESBL producers in urine specimens.

## 2. Materials and Methods

### 2.1. Study Design

In this study, we evaluated the performance of the ESBL NP test and the CTX-M LFA using clinical midstream urine (MSU) samples. The results obtained from the two tests were compared against phenotypic detection of ESBLs and PCR detection of the *bla*_CTX-M_ gene as the references (Figure 1). Testing was performed by a trained laboratory technologist in a clinical microbiology laboratory of a regional hospital (Kwong Wah Hospital) with 1400 beds in Hong Kong between November 2022 and January 2023. For the testing, 277 clinical MSU samples from unique patients aged ≥ 18 years were included.

In 60 MSU samples, direct testing was retrospectively performed on Day 3 after the bacterial identification and susceptibility testing results were available (Figure 1). Monomicrobial MSU samples positive for members of Enterobacterales (30 ESBL producers and 30 ESBL nonproducers) at ≥10^5^ CFU/mL were included in this part of the study. Polymicrobial MSUs or those positive for non-Enterobacterales were excluded. These retrospectively tested samples were tested prior to the prospective part of the study to validate the sample preparation protocols and to allow investigation of more samples with Enterobacterales. In another 217 prospective, clinical MSU samples, direct testing was performed on the day of specimen arrival (Day 1) after urine microscopic examination and before culturing was carried out. The inclusion criteria were (a) ≥ 50 white blood cells (WBCs) per five total fields (100× magnification, equivalent to ≥10 cells/μL) using an inverted microscope and (b) > 20 Gram-negative bacilli per 20 total fields (1000× magnification, equivalent to ≥10^5^ CFU/mL) in uncentrifuged urine samples [15,16]. Samples with mixed Gram-positive and Gram-negative bacteria on microscopic examination were excluded. All the bacterial isolates from the MSU samples were identified to the species level with susceptibility testing and detection of ESBL production as described below. All the Enterobacterales and other Gram negative bacterial isolates were further tested for the *bla*_CTX-M_ gene using CTX-M subgroup-specific multiplex polymerase chain reaction (PCR) assays as previously described [14,17].

### 2.2. Microbiological Methods

During the study period, containers with boric acid (0.4–0.6 g each) (Alleva Medical, Hong Kong, China) were provided to users for collection of MSU samples. The presence of boric acid helps to preserve the urine sample and prevent bacterial overgrowth during transport to the laboratory. On Day 1, dipsticks (Combur-3 Test, Roche, Basel, Switzerland) were used to measure pH, glucose and protein in the urine samples. The urine samples were plated quantitatively on CHROMID CPS Elite agar (bioMérieux, Marcy-l’Étoile, France), and the results were read after incubation for 16–20 h at 37 °C in an aerobic atmosphere. The result was considered positive if there was detection of one or more bacteria at a count of ≥10^5^ CFU/mL. On Day 2, bacterial identification was obtained using matrix-assisted laser desorption/ionization time-of-flight mass spectrometry (MALDI-TOF MS) with the Bruker Microflex^®^ LT Biotyper following the manufacturer’s instructions [18]. Susceptibility testing was performed using the disc diffusion method, and the results were interpreted according to the Clinical and Laboratory Standards Institute [19]. Phenotypic detection of ESBL production in Enterobacterales was performed using the combined disc test [19]. Two pairs of discs containing cefotaxime (30 μg) and cefotaxime-clavulanate (30/10 μg), ceftazidime (30 μg) and ceftazidime-clavulanate (30/10 μg) were placed onto an agar plate inoculated with the test organism and incubated at 37 °C for 16–18 h. ESBL production was interpreted to be positive if there was a ≥5 mm increase in the zone diameter for either cefotaxime or ceftazidime tested in combination with clavulanate, compared with the agent tested alone.

### 2.3. Methods for Assessing the Performance of the NG-Test CTX-M MULTI and the Rapid ESBL NP Test

For the CTX-M LFA, 10 mL of urine was centrifuged at 3000× *g* for 15 min; then, the supernatant was decanted (Figure 1). This was followed by mixing 150 µL of the sediment with 150 µL of the NG-Test extraction buffer and incubating the sample at 37 °C for 20 min for bacterial cell lysis [14]. Afterward, the sample well of the immunochromatographic cassette was loaded with 100 µL of the mixture, and the results were interpreted after 5 min and 15 min [12]. If there was only one band at the control region, the result was interpreted as negative. If there were two bands, one each at the control and test regions, the result was interpreted as positive.

For the ESBL NP test, 20 mL of urine was centrifuged at 3000× *g* for 15 min; then, the supernatant was decanted (Figure 1). The sediment was then washed twice using 20 mL distilled water. This involved resuspension of the sediment in 20 mL distilled water, centrifugation at 3000× *g* for 15 min and decanting to obtain the sediment. The final cell pellet was obtained by centrifugation of the sediment at 12,000× *g* for 2 min, and then 400 µL of lysis buffer was added; after 15 min, 100 µL of the solution was dispensed into each well in the panel: well a (no antibiotic), well b (cefotaxime) and well c (cefotaxime plus the ESBL inhibitor tazobactam). The inoculated panel was then incubated at 37 °C, and the results were interpreted after 20 min [11]. If wells a and c remained red and well b turned orange/yellow, the result was considered to be positive for ESBL phenotype. If well a remained red and wells b and c turned orange/yellow, the result was considered to be positive for cephalosporinase ± ESBL or carbapenemase ± cephalosporinase ± ESBL. If all wells (a, b, c) remained red, the result was negative for ESBL. If well a turned yellow or all the wells appeared yellow/orange, the result was considered invalid (i.e., noninterpretable).

### 2.4. Data Analysis

Data are presented as numbers and percentages (%). The results from the CTX-M LFA and the ESBL NP test were compared against phenotypic detection of ESBL and molecular detection of *bla*_CTX-M_ as reference methods [14]. The MedCalc statistical software package version 22.014 (MedCalc Software Limited, Ostend, Belgium) was used to calculate the sensitivity and specificity with a 95% confidence interval (CI).

## 3. Results

A total of 277 MSU samples were tested. In 60 MSU samples, direct testing was retrospectively performed two days after the identification of the organism, and susceptibility results were obtained. In 217 MSU samples, direct testing was performed immediately after screening. The urinalysis results indicated an acidic urinary pH (pH 5–6) in 89.5% of samples. The proportions of samples with glucose, protein, RBCs and WBCs were 9.7%, 56.7%, 21.9% and 97.1%, respectively (Table 1). In 270 samples, significant growth (≥10^5^ CFU/mL) of at least one Gram-negative bacillus organism was detected. In the remaining seven samples, a Gram-negative bacillus organism was present at 10^4^–10^5^ CFU/mL. The culture result was monomicrobial in 84.5% and polymicrobial in 15.5% of the samples. Among a total of 310 isolates, 15 different Gram-negative bacterial species were detected (Table 1). In 98.9% of the MSU samples, at least one Enterobacterales was detected. The most common organisms were *Escherichia coli* (78.3%), *Klebsiella pneumoniae* (14.4%) and *Proteus mirabilis* (8.7%).

Among the 304 isolates belonging to Enterobacterales, 75 (24.7%) were nonsusceptible to ceftazidime and/or ceftriaxone and 69 (22.6%) isolates exhibited an ESBL phenotype in the combined disc test. The number of ESBL producers tested in the retrospective and prospective MSU samples were 30 and 39, respectively. In the prospective study, ESBL-producing Enterobacterales was detected in 17.1% of the samples, including 22.2% in those aged ≥ 75 years, 11.8% in 51–74 years and 8.3% in 18–50 years. Among the *E. coli* and other Enterobacterales isolates, ESBL prevalence was 20.1% and 5.7%, respectively. All 69 ESBL-positive isolates (from 67 MSU samples) were PCR-positive for the *bla*_CTX-M_ gene. The isolates were positive for the CTX-M-1 subgroup (*n* = 26), the CTX-M-9 subgroup (*n* = 40), both the CTX-M-1 and CTX-M-9 subgroups (*n* = 2) and the CTX-M hybrid subgroup (*n* = 1). The other *Enterobacterales* isolates and those of other bacterial groups were PCR-negative for the *bla*_CTX-M_ gene (Table 1).

The CTX-M LFA yielded true-positive results in all 67 urine samples with organisms that produce CTX-M. These included 63 urine samples (51 monomicrobial and 12 polymicrobial) with CTX-M producers at 10^5^ CFU/mL and 4 urine samples (3 monomicrobial and 1 polymicrobial) with CTX-M producers at 10^4^–10^5^ CFU/mL. The CTX-M producers (*n* = 69) recovered from the samples included *E. coli* (*n* = 62), *K. pneumoniae*, (*n* = 3), *P. mirabilis* (*n* = 3) and *Enterobacter cloacae* (*n* = 1). In the 210 urine samples that did not contain CTX-M producers, CTX-M LFA yielded two false-positive results (1 monomicrobial sample with 1 *E. coli* isolate and 1 polymicrobial sample with 2 *E. coli* isolates with different resistance patterns) and 208 true-negative results (179 monomicrobial and 29 polymicrobial). The three *E. coli* isolates from the two samples with false-positive results were ESBL-negative by the combined disc test, and PCR results for the *bla*_CTX-M_ gene were negative as well. All three culture isolates were further tested using CTX-M LFA, and negative results were obtained (Figure 2). Considering all the urine samples, the CTX-M LFA showed 100% (95% CI, 94.6–100%) sensitivity and 99.1% (95% CI, 96.6–99.9%) specificity (Table 2). The sensitivity and specificity were similar for the MSU samples tested retrospectively and immediately after urinalysis. In the samples with positive results, the band at the test region was apparent at 5 min following sample loading, and no obvious difference was observed at 15 min.

The ESBL NP test yielded interpretable results in 274 of the 277 samples (Table 2). These included 58 true-positives, 207 true-negatives and 9 false-negatives. The results for three urine samples were invalid (Figure 3). No false-positive result was observed. None of the ESBL NP test results indicated the presence of cephalosporinase. This test failed to detect ESBL producers (seven *E. coli*, two *P. mirabilis*) in nine urine samples (seven monomicrobial and two polymicrobial) (Figure 3). The ESBL producers had counts of 10^5^ CFU/mL in five samples and 10^4^ to 10^5^ CFU/mL in four samples. The culture isolates of the nine ESBL producers were further tested using ESBL NP tests, and positive results were obtained. The overall sensitivity and specificity of the ESBL NP test was 86.6% (95% CI, 76.0–93.7%) and 100% (95% CI, 98.2–100%), respectively.

## 4. Discussion

We presented data on the comparative performance of CTXM LFA and ESBL NP tests for the direct detection of CTX-M producers in MSU samples. The data showed that both tests can be successfully performed with urine samples containing boric acid. This has practical significance because boric acid is commonly added as a preservative to prevent bacterial overgrowth during transport to the laboratory [20]. In the ESBL NP test, the hydrolysis of cefotaxime by ESBL decreases the pH of the medium, which causes the phenol red in the reagent of the test to turn yellow [11]. Consequently, the presence of boric acid is expected to interfere with the performance of the ESBL NP test due to acidification of urine. This explains why a large proportion of the urine samples had pH values between five and six. Therefore, the sample preparation for the ESBL NP test in this study involved washing steps to remove boric acid. Our pilot testing found that two washing steps were required to reliably mitigate the interference from boric acid. Invalid results occurred in three urine samples with the ESBL NP test, giving an overall invalidity rate of ~1%, which is acceptable. The small number of invalid results might be due to other acidic substances left in the urine sediments. Since an acidic pH may inhibit the antigen–antibody reaction, the presence of boric acid may cause invalid results or decrease the sensitivity of the CTX-M LFA [21]. In the present study, all the CTX-M LFAs were performed without any wash step, and all the results were interpretable. Thus, the CTX-M LFA is technically easier to perform than the ESBL NP test. Furthermore, the CTX-M LFA had higher sensitivity than the ESBL NP test, while both tests had very high specificities. By comparison, the CTX-M LFA had a 91.6–100% sensitivity and a 95.0–100% specificity for direct testing of positive blood culture broths [13,14,22,23,24,25,26,27]. In our earlier study that used the CTX-M LFA to test positive clinical blood culture broths, the sensitivity and specificity were 100% (95% CI, 96.0–100%) and 99.6% (97.9–100%), respectively [14]. In a study that investigated 142 blood cultures positive for *K. pneumoniae* or *E. coli*, the CTX-M LFA and the ESBL NP test were found to perform equally with a sensitivity of 92% (95% CI, 77–98%) and a specificity of 92% (95% CI, 77–98%), respectively [13].

In this study, all the urine samples with ESBL-producing Enterobacterales were correctly identified with the CTX-M LFA. The finding should be interpreted in the context that 100% of the ESBL producers were CTX-M producers. In the prospective part of this study, a relatively high proportion (17.1%) of the urine samples were positive for ESBL-producing Enterobacterales. This rate is broadly similar to the rates obtained in recent surveys of UTIs in the Asia-Pacific region such as 20.2% in 2016 for Bhutan, 33.8% in 2018 for the Republic of Korea and 21.6% in 2017 for Thailand [28]. In Hong Kong, CTX-M is the predominant type of ESBL detected in clinical isolates of *E. coli*, *K. pneumoniae*, *E. cloacae* and *Proteus mirabilis* [7,29,30]. In our previous study, CTX-M was detected in all 90 ESBL-producing blood culture isolates of Enterobacterales from two regional hospitals during 2021–2022 [14]. In our locality, CTX-M-14 and CTX-M-15 are the most common alleles, and ST131 and ST405 are the two major *E. coli* clones associated with bloodstream infection and UTIs [3,6,8]. In areas where non-CTX-M ESBLs are prevalent, the ESBL NP test may be preferred over the CTX-M LFA [4]. In the samples with CTX-M producers, the CTX-M band was of an intensity comparable to the control band and was apparent within 5 min after loading the cassette. In contrast, the two samples with false-positive CTX-M LFA results had a weak CTX-M band relative to the control band. We acknowledge that a weak band may be variably interpreted by different observers as invalid or a false-positive. In an earlier study in which the CTX-M LFA was performed on cell pellets obtained from 37 urine samples, no false-positive results were observed [31]. In direct testing of blood cultures positive for Gram-negative bacilli, a small number of false-positive CTX-M LFA results with a weak band have been previously reported [14,27]. In our earlier study of 220 clinical blood cultures, one false-positive result was observed [14]. In another study of 167 clinical blood cultures, two false-positive results were reported [27].

In the present study, urine samples were included for direct testing using a 2-tier screening for pyuria and presence of Gram-negative bacilli in a Gram stain. Previous studies have shown that dipstick leukocyte esterase is not a sensitive indicator for detection of pyuria and could fail to identify up to 60% of significant pyuria (≥10 WBC/μL) detected using microscopic examination [32,33]. This is why we used inverted microscopy as our first-tier screening. In many laboratories, urinalysis parameters (e.g., presence of WBC and/or bacteria) are adopted as indicators for reflex urine culture approaches to improve cost-effectiveness and implement antimicrobial stewardship [34,35]. In modelling analysis, absence of pyuria was found to have a very high negative predictive value for significant bacteriuria [36]. We used the presence of Gram-negative bacilli in uncentrifuged urine (≥20 organisms per 20 microscopic fields) as a surrogate indicator for performance of the two ESBL detection tests. Comparison with subsequent culture results showed that this is highly predictive of the presence of Enterobacterales at significant counts (≥10^5^ CFU/mL). Our findings are consistent with previous studies evaluating the usefulness of the Gram stain as an aid for the diagnosis of UTIs [37,38]. It should be pointed out that manual microscopic examination of urine samples is relatively labor-intensive. Consequently, automated flow cytometry-based analyzers have been introduced to allow for a more robust screening of urine samples for WBC counts and bacterial load differentiation [39,40]. Although negative results from these analyzers have yielded excellent negative predictive value and could streamline the workflow, the specificities and positive predictive values for UTI diagnosis have been reported to be suboptimal, and their ability to discriminate between Gram-negative and Gram-positive UTIs is low [41,42,43]. In our locality, a recent study found that the concordance rate between a flow cytometry analyzer and culture results for predicting Gram-positive or Gram-negative UTIs was 38% [41].

This study has several limitations. First, both rapid tests only target one resistance mechanism; thus, a negative test result cannot be used to rule out extended-spectrum cephalosporin resistance. In the current study, 6 out of 75 cefotaxime and/or ceftazidime-resistant Enterobacterales isolates were negative in both tests. Second, the detection of non-CTX-M-type ESBL producers with the ESBL NP test was not evaluated. Third, the manufacturer’s instructions specified that both tests are intended to be performed with culture colonies [11,12]. Despite promising results from the current and previous studies, both tests have not been approved for direct testing of clinical specimens [13,14,31]. Approval of diagnostic tests for direct performance on clinical specimens would need application by the manufacturers concerned and fulfillment of regulatory requirements. We did not use MALDI-TOF MS on urine samples directly for bacterial identification, as the accuracy and concordance with culture results have been reported to be unsatisfactory [31,44]. In one study, which performed direct MALDI-TOF MS on urine sediment when a single morphotype was detected by direct Gram staining, correct identification was obtained in 137 (66.2%) of 207 samples [31]. In another study that screened urine samples with flow cytometry, concordant direct identification was obtained in 54 (69.2%) of 78 urine samples with significant bacterial growth (44). The main advantage of our test approach is that ESBL results are available on Day 1 before culture. The information can be valuable for patients who are hospitalized for upper UTIs and receiving empirical therapy with ceftriaxone or piperacillin-tazobactam, as these antibiotics are ineffective or suboptimal for UTIs caused by ESBL-producing Enterobacterales [45]. We acknowledge that a rapid test for ESBL production can be performed on Day 2 using colonies from agar plates and after bacterial identification is obtained with MALDI-TOF MS. The disadvantage of such an approach, as with our approach, is that the ESBL results would be available one day later. For direct testing of urine samples, the present study provides a comprehensive, comparative analysis of two ESBL tests against molecular detection as the reference method. The 2-tier urine screening workflow, parallel assessment of the two tests and a relatively large sample size are the present study’s strengths.

## 5. Conclusions

This study showed that rapid detection of CTX-M producers directly in MSU samples can be achieved using two tests previously validated for ESBL detection in culture isolates. The higher sensitivity of the CTX-M LFA and its simple sample preparation requirement make it a preferred test over the ESBL NP test in areas where CTX-M is prevalent. The implementation of these rapid tests in clinical laboratories provides an opportunity for early optimization of antibiotic administration in patients with UTIs.

## Figures and Tables

**Figure 1 antibiotics-12-01585-f001:**
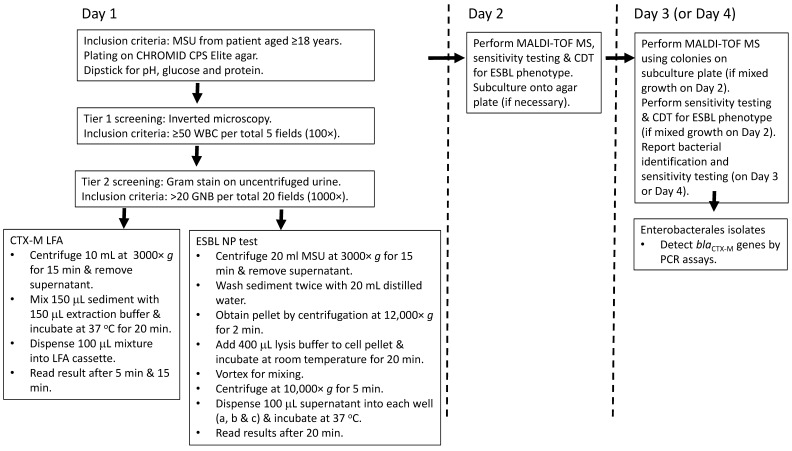
Flow chart showing the diagnostic steps in the study. Abbreviations: CDT, combined disc test; ESBL, extended-spectrum β-lactamase; GNB, Gram-negative bacilli; LFA, lateral flow assay; MALDI-TOF MS, matrix-assisted laser desorption/ionization time-of-flight mass spectrometry; MSU, midstream urine; WBC, white blood cells.

**Figure 2 antibiotics-12-01585-f002:**
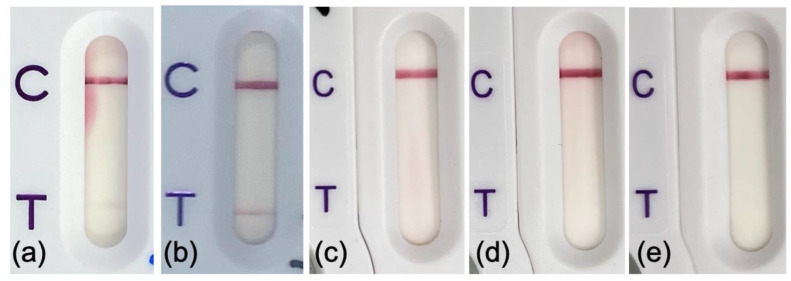
NG-Test CTX-M MULTI results obtained for two urine samples and bacterial colonies of isolates from the samples. A weak CTX-M band (false-positive) was observed in direct testing of the two urine samples, (**a**) urine P051 and (**b**) urine P193. Negative results were obtained for the culture colonies of (**c**) *Escherichia coli* from P051, (**d**) *E. coli* 1 from P193 and (**e**) *E. coli* 2 from P193. Sample P193 was culture-positive for two different *E. coli*. C stands for the control line, and T for the CTX-M test line.

**Figure 3 antibiotics-12-01585-f003:**
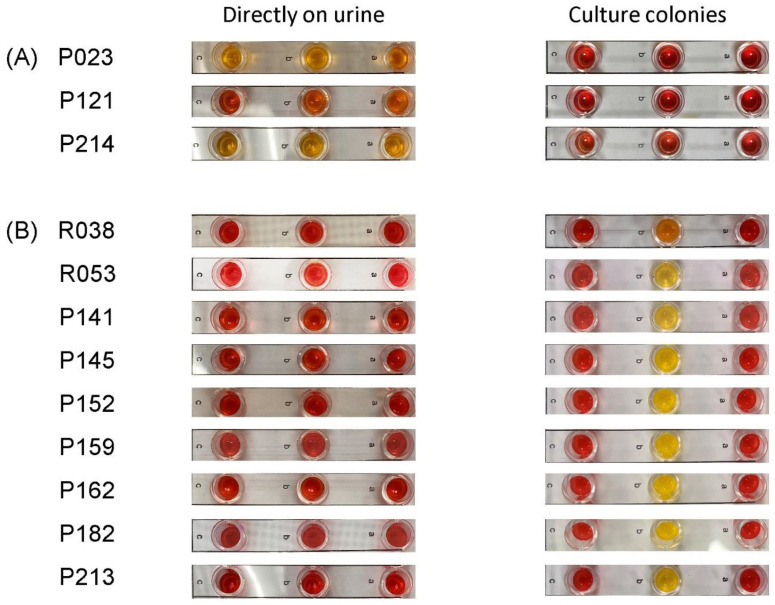
Rapid ESBL NP test showing (**A**) 3 invalid results (all wells orange) for testing performed on urine samples directly, (**B**) 9 negative results (all wells red) for testing performed on urine samples directly and 9 positive results (well A and well C red, well B orange/yellow) for testing performed on culture colonies from the corresponding samples. The label on the left shows the urine sample number. The letters in the strips indicate the wells with no antibiotic (a), cefotaxime (b) and cefotaxime plus tazobactam (c).

**Table 1 antibiotics-12-01585-t001:** Summary of urine samples tested in this study.

Characteristic	No. (%) of Urine Samples
Urinalysis (concentration range)	
pH 5–6	248 (89.5)
pH 7–8	29 (10.5)
Glucose + (2.8–56 mmol/L)	27 (9.7)
Glucose −	250 (90.3)
Protein + (0.3–5 g/L)	157 (56.7)
Protein −	120 (43.3)
RBC + (1– > 100/µL)	58 (21.9)
RBC −	219 (79.1)
WBC + (10– > 100/µL)	269 (97.1)
WBC −	8 (2.9) ^c^
Samples with organisms	
*Escherichia coli*	217 (78.3)
*Klebsiella pneumoniae*	40 (14.4)
*Proteus mirabilis*	24 (8.7)
Other Enterobacterales ^a^	23 (8.3)
*Acinetobacter baumannii* complex	4 (1.4)
*Pseudomonas aeruginosa*	2 (0.7)
Other bacteria ^b^	13 (4.7)
Number of organisms	
Monomicrobial	234 (84.5)
Polymicrobial	43 (15.5)
CTX-M identified	
CTX-M-1 subgroup	24 (8.7)
CTX-M-9 subgroup	40 (14.4)
Both CTX-M-1 and CTX-M-9 subgroup	2 (0.7)
CTX-M hybrid subgroup	1 (0.4)
Negative for CTX-M	210 (75.8)
Total	277 (100)

^a^ Including 8 Citrobacter koseri, 3 Morganella morganii, 3 Providencia stuartii, 2 K. oxytoca, 2 Serratia marcescens and 1 each of Enterobacter cloacae, E. hormaechei, Klebsiella aerogenes, Klebsiella variicola and Proteus vulgaris. ^b^ Including 7 Enterococcus faecalis, 2 Streptococcus agalactiae and 1 each of Corynebacterium striatum, Streptococcus dysgalactiae, Streptococcus gallolyticus and Streptococcus vestibularis. ^c^ All samples tested prospectively were WBC +. The samples with a WBC − result were tested retrospectively. +, positive; −, negative.

**Table 2 antibiotics-12-01585-t002:** Performance of rapid methods for direct detection of ESBL-producing Enterobacterales in midstream urine samples.

Method and Sample Description	Result (No.)				Test Performance ^d^	
	**TP**	**FP**	**FN**	**TN**	**Sensitivity, %**	**Specificity, %**
NG CTX-M MULTI						
Retrospective testing ^a^	30	0	0	30	100 (88.4–100)	100 (88.4–100)
Immediate testing ^b^	37	2	0	178	100 (90.5–100)	98.9 (96.0–99.9)
Total	67	2	0	208	100 (94.6–100)	99.1 (96.6–99.9)
Rapid ESBL NP						
Retrospective testing ^a^	28	0	2	30	93.3 (77.9–99.2)	100 (88.4–100)
Immediate testing ^b^	30	0	7	177 ^c^	81.1 (64.8–92.0)	100 (97.9–100)
Total	58	0	9	207	86.6 (76.0–93.7)	100 (98.2–100)

TP, true-positive; FP, false-positive; FN, false-negative; TN, true-negative. ^a^ Testing performed on 60 urine samples following bacterial identification. ^b^ Testing performed on 217 urine samples immediately after urinalysis. ^c^ Results were invalid for 3 urine samples with ESBL-nonproducing Enterobacterales. The 3 samples were excluded in the sensitivity and specificity analysis. ^d^ 95% confidence interval given inside brackets.

## Data Availability

All relevant data have been included in this manuscript.

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
