# Peer review of "Evaluation of Two Tests for the Rapid Detection of CTX-M Producers Directly in Urine Samples"

_antibiotics, 2023, doi:10.3390/antibiotics12111585_

Round 1

Reviewer 1 Report

Comments and Suggestions for Authors

The manuscript by Tang and colleagues compares 2 rapid detection methods to detect CTX-M producers in urine samples. The study is of medical relevance and the manuscript is well-written, with well-designed experiments and clearly demonstrated results. The work presents important information that can be used to improve clinical methodologies in healthcare worldwide. 

However, I do have some comments and suggestions that I think should be addressed before it is accepted for publication. 

1) A general schematic organigram of the study would be very helpful to better elucidate the steps of the study design. 

2) As properly discussed by the authors in the discussion part, they should mention in the "results" that boric acid was used in samples preparation (and therefore decreased urine pH). Complementarily, there is no mention of how the acidification affects the tests directly (i.e. does it affect the reaction or stability of reagents), please include a short explanation to justify this.  

3) While comparing with other previous studies, the obtained results from the referred works could be included in the text, to facilitate the direct comparison of the reader. Examples: row 167 (mention surveys rates); row 178-179 (mention the rates obtained in the referred works). 

4) Row 209: Authors can better elaborate and discuss why the tests used have not yet been approved for direct testing clinical species (is it due to sensitivity issues, limitations with the reaction interferences, etc..).

In order to increase the originality of the work, a more in-depth comparative analysis could be made by the authors by comparing the previous work performed with blood samples in the same locality (and also with other studies that tested direct urine and blood tests). In terms of comparing the prevalence results, tests sensitivity and specificities, species found, etc.  This is just a suggestion of comparison, but authors can include other relevant information to the comparison in order to enrich the discussion. Otherwise, the work could sound very repetitive when compared to Microorganism 11(1), 128 - 2023 - published by the same group (same tests performed, only changed the source of samples - which can be very important in clinical practical aspects, but scientifically poor).  

 5) the methodology part could be more comprehensive. Please consider subdividing in topics according to the specific technical approach used. 

Author Response

Reviewer 1

Comments and Suggestions for Authors

The manuscript by Tang and colleagues compares 2 rapid detection methods to detect CTX-M producers in urine samples. The study is of medical relevance and the manuscript is well-written, with well-designed experiments and clearly demonstrated results. The work presents important information that can be used to improve clinical methodologies in healthcare worldwide. 

RESPONSE

Thank you for reviewing our manuscript and the comments for improvement.

However, I do have some comments and suggestions that I think should be addressed before it is accepted for publication. 

1) A general schematic organigram of the study would be very helpful to better elucidate the steps of the study design. 

RESPONSE

We have added a Figure to illustrate the study design and workflow.

2) As properly discussed by the authors in the discussion part, they should mention in the "results" that boric acid was used in samples preparation (and therefore decreased urine pH). Complementarily, there is no mention of how the acidification affects the tests directly (i.e. does it affect the reaction or stability of reagents), please include a short explanation to justify this.  

RESPONSE

We have further explained the purpose of using urine container with boric acid in the method section. The boric acid containing specimen collection bottles were commercially supplied. The presence of boric acid helps to preserve the urine sample and prevent bacterial overgrowth during transport to the laboratory. Boric acid was not added during any steps in the sample preparation for performance of the two tests. Hence, we clarified this in the method section and this part was not described as “results”.

In the Discussion, we added explanation on how boric acid might affect the two tests.

3) While comparing with other previous studies, the obtained results from the referred works could be included in the text, to facilitate the direct comparison of the reader. Examples: row 167 (mention surveys rates); row 178-179 (mention the rates obtained in the referred works). 

RESPONSE

Thank you for the suggestion. We have added the rates from the referred works to those sentences in the Discussion section.

4) Row 209: Authors can better elaborate and discuss why the tests used have not yet been approved for direct testing clinical species (is it due to sensitivity issues, limitations with the reaction interferences, etc..).

RESPONSE

For purpose of regulatory approval, the manufacturer would need to make application and provide all the necessary studies for the intended use of the test in accordance with the regulatory requirements. The requirements are agency specific and may involve testing for specified number of samples for reproducibility, interference, comparison with an approved test and other requirements. There could be costs or other consideration. We believe that it is not appropriate for us to speculate why the two manufacturers have not sought approval from the regulatory agencies. We added a sentence in the Discussion to address this.

In order to increase the originality of the work, a more in-depth comparative analysis could be made by the authors by comparing the previous work performed with blood samples in the same locality (and also with other studies that tested direct urine and blood tests). In terms of comparing the prevalence results, tests sensitivity and specificities, species found, etc.  This is just a suggestion of comparison, but authors can include other relevant information to the comparison in order to enrich the discussion. Otherwise, the work could sound very repetitive when compared to Microorganism 11(1), 128 - 2023 - published by the same group (same tests performed, only changed the source of samples - which can be very important in clinical practical aspects, but scientifically poor).  

RESPONSE

The current work is very different from the one we reported earlier in Microorganism 11(1), 128 – 2023. The two pieces of work are obviously not repetitive. First, blood culture broths were tested after they were culture positive for Gram negative bacillus. Here, urine samples were tested directly before culture. Second, only the LFA test was evaluated in our blood culture study. Here, we compared the performance of two tests for direct testing of urine samples. We address this comment by expanding  the text on comparison with the previous work performed with blood culture samples in the Discussion section. 

 5) the methodology part could be more comprehensive. Please consider subdividing in topics according to the specific technical approach used. 

RESPONSE

The method section has been extensively reorganized and rewritten. New subheadings have been added to organize the content according to the technical approach. A Figure has been added to illustrate the workflow.

Reviewer 2 Report

Comments and Suggestions for Authors

Tang F et al explore an interesting topic - rapid detection of CTX-M/ESBL production directly in urine sample, the results and the problem are discussed very well. The paper is worth to publish but some questions arisen.

-          Please include keywords

-          Line 67  - please include that “ the proportion of samples with elevated amounts of glucose, protein, RBC and WBC (references are shown in Table 1) was …..

Line 235 – Does the combination disk test apply to all microorganisms detected?  CLSI recommends using this method only for K pneumoniae, E. coli, and P mirabilis.

Table 1 please write Acinetobacter spp and Pseudomonas spp. Does this mean that MALDI Toff can not recognize these isolates at the species level?

Table 1 Please write  “sample with:” instead of “number of organisms”; please write CTX-M-1 and CTX-M-9 instead of M1 and M2

Information on the spectrum of ESBL enzymes in Hong Kong could be included in the discussion section, this would provide important information on the utility of the proposed rapid test for detection of CTX-M.

Another point of view is –identification of bacteria is important for prescription of appropriate therapy, MALDI Toff use cultured isolates. In this case why to use directly urine, but not culture for rapid tests? Please include some discussion of this issue.

Author Response

Reviewer 2

Comments and Suggestions for Authors

Tang F et al explore an interesting topic - rapid detection of CTX-M/ESBL production directly in urine sample, the results and the problem are discussed very well. The paper is worth to publish but some questions arisen.

-          Please include keywords

RESPONSE

Six keywords have been included.

-          Line 67  - please include that “ the proportion of samples with elevated amounts of glucose, protein, RBC and WBC (references are shown in Table 1) was …..

RESPONSE

We have cited Table 1 against this sentence according to this suggestion.

Line 235 – Does the combination disk test apply to all microorganisms detected?  CLSI recommends using this method only for K pneumoniae, E. coli, and P mirabilis.

RESPONSE

All Enterobacterales isolates were tested for ESBL production using the CDT. This has been clarified. As described in the results, a blaCTX-M positive Enterobacter cloacae isolate was correctly detected by the CDT.

Table 1 please write Acinetobacter spp and Pseudomonas spp. Does this mean that MALDI Toff can not recognize these isolates at the species level?

RESPONSE

These isolates were identified as Acinetobacter baumannii complex and Pseudomonas aeruginosa. The names in Table 1 have been revised accordingly.

Table 1 Please write  “sample with:” instead of “number of organisms”; please write CTX-M-1 and CTX-M-9 instead of M1 and M2

RESPONSE

They have been revised in Table 1.

Information on the spectrum of ESBL enzymes in Hong Kong could be included in the discussion section, this would provide important information on the utility of the proposed rapid test for detection of CTX-M.

RESPONSE

Thank you for the suggestion. We have added the information to the Discussion section.

Another point of view is –identification of bacteria is important for prescription of appropriate therapy, MALDI Toff use cultured isolates. In this case why to use directly urine, but not culture for rapid tests? Please include some discussion of this issue.

RESPONSE

Thank you for the suggestion. Discussion on this issue has been added.

Round 2

Reviewer 1 Report

Comments and Suggestions for Authors

The authors have replied accordingly, edited the text, and inserted relevant information that was missing. I consider the manuscript ready for publication.